# Peroral Pancreatoscopy-Guided Lithotripsy Compared with Extracorporeal Shock Wave Lithotripsy in the Management of Pancreatic Duct Stones in Chronic Pancreatitis: A Multicenter Retrospective Cohort Study

**DOI:** 10.3390/diagnostics14090891

**Published:** 2024-04-24

**Authors:** Keisuke Iwata, Takuji Iwashita, Tsuyoshi Mukai, Yuhei Iwasa, Mitsuru Okuno, Kensaku Yoshida, Akinori Maruta, Shinya Uemura, Ichiro Yasuda, Masahito Shimizu

**Affiliations:** 1Department of Gastroenterology, Gifu Municipal Hospital, Gifu 500-8513, Japan; 2First Department of Internal Medicine, Gifu University, Gifu 501-1194, Japan; 3Department of Gastroenterological Endoscopy, Kanazawa Medical University, Ishikawa 920-0293, Japan; 4Department of Gastroenterology, Gifu Prefectural General Medical Center, Gifu 500-8717, Japan; 5Third Department of Internal Medicine, Graduate School of Medicine University of Toyama, Toyama 930-0194, Japan

**Keywords:** pancreatic duct stone, chronic pancreatitis, peroral pancreatoscopy (POPS), extracorporeal shock wave lithotripsy (ESWL), endoscopic therapy

## Abstract

Background: Extracorporeal shock wave lithotripsy (ESWL) is a common treatment for pancreatic stones in chronic pancreatitis. In contrast, peroral pancreatoscopy-guided lithotripsy (POPS-L) remains underexplored, with limited comparative studies to ESWL. This study compared the treatment outcomes of disposable POPS-L tools and ESWL for pancreatic stones. Methods: A retrospective analysis was conducted on 66 patients who had undergone pancreatic stone treatment at three institutions between 2006 and 2022. The treatment outcomes of POPS-L and ESWL were compared. Results: This study included 19 and 47 patients who had undergone POPS-L and ESWL, respectively. In a comparison between POPS-L and ESWL, the stone clearance rates were 78.9% vs. 70.2% (*p* = 0.55), while the procedure-related complication rates were 21% vs. 6.3% (*p* = 0.09). The median total session counts were 1 vs. 5 (*p* < 0.01). The cumulative stone recurrence rates were comparable in both groups. Multivariate analysis revealed no significant factors influencing the stone clearance rates, and the choice between POPS-L and ESWL did not affect the stone clearance rates. Conclusions: POPS-L and ESWL exhibited comparable treatment outcomes in terms of stone clearance, complications, and recurrence rates. Furthermore, POPS-L is advantageous due to the need for fewer sessions to achieve pancreatic stone clearance.

## 1. Introduction

Pancreatic duct stones resulting from chronic pancreatitis may lead to symptoms such as abdominal and back pain, the acute exacerbation of pancreatitis, pseudocyst formation, and cyst infection, which may necessitate prompt intervention. Furthermore, the obstruction of pancreatic fluid drainage by these stones may lead to pancreatic parenchymal atrophy, causing irreversible endocrine and exocrine dysfunction and an increased risk of nutritional disorders, including diabetes and malabsorption [1,2]. Additionally, the risk of pancreatic cancer may increase with the persistence or exacerbation of chronic pancreatitis [3]. Traditional surgical interventions have long been employed to treat pancreatic stones, demonstrating favorable shortened long-term outcomes; however, their invasiveness is concerning [4,5]. 

Following the initial report on the effectiveness of extracorporeal shock wave lithotripsy (ESWL) in 1987 [6], ESWL has been widely used as a minimally invasive treatment for pancreatic stone management with promising results in terms of stone fragmentation, stone removal, symptom resolution rates [7], and long-term pain relief rates [8]. Nevertheless, a few patients may present challenges in achieving sufficient stone fragmentation even with the use of ESWL or difficulties due to pain during treatment. Other challenges include maintaining the required body position, leading to patients in whom treatment implementation is hindered. 

The utility of peroral pancreatoscopy-guided lithotripsy (POPS-L), which involves techniques such as laser lithotripsy (LL) or electrohydraulic lithotripsy (EHL) directly under peroral pancreatoscopy (POPS), has been previously reported [9]. Recently, the improved operability of disposable POPS tools has emerged, suggesting its potential to enhance treatment outcomes in the management of pancreatic stones. However, in the management of pancreatic stones, the utility and safety of these advances have not been sufficiently investigated. Therefore, this novel study compared the effectiveness and safety of disposable POPS-L tools and ESWL for the treatment of pancreatic stones. 

## 2. Materials and Methods

### 2.1. Patients

This multicenter retrospective cohort study was conducted at Gifu Municipal Hospital, Gifu University Hospital, and Gifu Prefectural General Medical Center. This involved the analysis of a database containing all patients with chronic pancreatitis in whom treatment for pancreatic stones had been performed between 2006 and 2022. The study included patients who met the following inclusion criteria: (1) patients who had undergone either ESWL or disposable-type POPS-L for pancreatic stones; (2) patients whose main pancreatic duct stone size had been ≥ 5 mm; and (3) patients whose symptoms had included issues such as abdominal pain, acute exacerbation of pancreatitis, obstructive jaundice, pseudocyst, or pancreatic pleural effusion, or those desiring treatment to preserve pancreatic function. The exclusion criteria of the study comprised patients with recurrent pancreatic stones. Regarding the choice between POPS-L and ESWL, ESWL had been selected until the availability of disposable-type POPS-L tools. After the availability of disposable-type POPS-L tools, both treatments had been explained to the patient, and the choice of treatment had been primarily determined by the patient’s preference.

This retrospective chart review study involving human participants was conducted in accordance with the ethical standards of the institutional and national research committee, as well as with the 1964 Helsinki Declaration and its subsequent amendments or comparable ethical standards. The Institutional Review Board of our facilities approved this study. Written informed consent for the procedures was obtained from all the participants; informed consent for this study was obtained using an opt-out approach.

### 2.2. ESWL Procedure

Before the ESWL, an endoscopic pancreatic duct sphincterotomy was performed in all the patients. Additionally, pancreatic duct stenting or naso-pancreatic drainage (NPD) tube placement was generally performed to prevent pancreatitis due to the impaction of fragmented pancreatic stones and facilitate visualization during the ESWL in patients with poorly visualized stones under fluoroscopy. The choice between EPS and NPD was made at the discretion of the operator. The ESWL was performed using a Lithotripter S II (Dornier, Lindau, Germany) or LITHOSTAR Multiline (Siemens, Munich, Germany). The patient was placed in a prone position, and approximately 3000–4000 shocks per session were administered, with multiple sessions performed until successful fragmentation was confirmed. The number of sessions was typically set at 2–3 per week. The fluoroscopic images obtained during the ESWL or CT scans were used to assess the status of the stone fragmentation. If endoscopic stone extraction was deemed necessary or effective based on the presence of residual fragments, endoscopic stone removal was employed using stone retrieval balloons or baskets. All the ERCP and ESWL procedures were performed on an in-patient basis. 

### 2.3. POPS-L Procedure

An endoscopic pancreatic duct sphincterotomy was performed before treatment or within the same session. Subsequently, a disposable POPS tool (Spyglass DS, Boston Scientific, United States) was inserted over a guidewire placed within the pancreatic duct. The POPS tool was advanced to face the pancreatic stone, and either electrohydraulic lithotripsy (EHL) using AUTOLITH (Northgate Technologies Inc., Illinois, United States) or laser lithotripsy (LL) using a holmium:YAG Laser (Lumenis, Versa Pulse Select 80 W, Yokneam, Israel) and a 200-µm laser fiber with energy settings of 0.5–1.0 J and a rate of 5–10 Hz were performed (Figure 1). The selection between EHL and LL adhered to the principle of primarily utilizing EHL; however, in instances where EHL was unavailable, LL was chosen. In patients where the insertion of the EHL or LL tools was challenging due to strong bends in the POPS tool, the POPS tool was removed from the ERCP scope. Subsequently, it was re-inserted into the pancreatic duct through the same ERCP scope, while pre-loaded with EHL/LL. Stone fragmentation was performed while ensuring adequate visualization through intermittent irrigation. Additionally, intermittent suction was used as needed to control the pancreatic duct pressure. After successful stone fragmentation, endoscopic stone removal was performed as required using stone retrieval balloons or baskets.

### 2.4. Evaluation Criteria

The primary evaluation criterion was the rate of clearance of the targeted main pancreatic duct stones. We also assessed the incidence of complications, treatment duration, total number of treatment sessions (including both endoscopic therapy and ESWL), cumulative recurrence rate, and factors influencing the stone clearance rate as secondary outcomes. 

### 2.5. Definitions

Complete clearance of the target pancreatic stones was defined as treatment success. Complete pancreatic stone clearance was determined during the final ERCP using fluoroscopic images or POPS. The total number of sessions included the sum of the endoscopic sessions involving EPST, EPS/ENPD, pancreatic duct dilation, pancreatic stone extraction, and ESWL. We determined pancreatic duct stenosis to be present when any treatment, such as balloon dilation or pancreatic stenting, was necessary to achieve complete stone clearance or relieve pain. Post-ERCP hyperamylasemia was defined as amylase at least three times the normal level at more than 24 h after the procedure, without clinical pancreatitis symptoms. Post-ERCP pancreatitis was defined as clinical pancreatitis accompanied by amylase at least three times the normal level at more than 24 h after the procedure. The recurrence of pancreatic stones was defined as the reappearance of stones on image findings, irrespective of symptoms. 

### 2.6. Statistical Analyses

For the nominal variables, Fisher’s Exact Test or Pearson’s chi-square test were used for comparison, as appropriate. The Wilcoxon sum-rank test was used to compare the continuous variables. The time to stone recurrence was estimated using the Kaplan–Meier method and compared using the log-rank test. The factors influencing treatment success were evaluated using univariate logistic regression analysis, and variables with a *p*-value of 0.2 or less, along with clinically relevant factors, were included in the multivariate analysis. The level of significance was set at a *p*-value ≤ 0.05. Statistical analyses were performed using JMP (version 11). 

## 3. Results

### 3.1. Characteristics of Patients and Pancreatic Stones

The POPS-L group (POPS group) consisted of 19 patients, whereas the ESWL group consisted of 47 patients. The median age of the patients was 64 years (range: 22–86 years), and 52 male patients were included in the study. The etiology of the pancreatitis was related to alcohol consumption in 58 patients, idiopathic in 7, and autoimmune in 1. The treatment indications were pain or acute exacerbation of pancreatitis in 53 patients, the management of pancreatic pseudocysts in 2, and the anticipation of preserving pancreatic function in 11 (Table 1). 

The median diameter of the targeted stones, median number of stones, locations of the stones, and median CT value were as follows: a size of 10 mm (range: 5–40 mm); 1 stone (range: 1–3 stones); in the head in 51 patients and in the body/tail in 15 patients; and a CT value of 1043 Hounsfield Units (HU) (range: 156–3814 HU). Pancreatic duct stenosis downstream of the pancreatic stone was observed in 28 patients (42%), with a median pancreatic duct diameter of 7.5 mm (range, 3–15 mm) on the tail side of the stone (Table 2). There were no significant differences between the POPS and ESWL groups in any of the background factors. 

### 3.2. Treatment Outcomes

Among the 19 patients in the POPS group, in 2 patients it was impossible to reach the pancreatic stone with the POPS tool due to significant bending or narrowing of the pancreatic duct (Table 3). Among the remaining 17 patients, 4 underwent laser lithotripsy (LL) and 13 underwent electrohydraulic lithotripsy (EHL). The median number of sessions was one (range: 1–4 sessions), and the median total procedural time was 72 min (range: 30–187 min). In the ESWL group of 47 patients, 17 patients underwent ESWL alone and 30 patients underwent ESWL in combination with endoscopic stone removal. The median number of ESWL sessions was 3 sessions (range: 1–22 sessions), with a median total number of shock waves of 9000 (range: 2000–70,000). For patients with stenosis of the pancreatic duct in the papilla side of the stone, dilation treatment (using 4 mm or 6 mm balloon or mechanical dilators, either singly or in combination) was performed in eight and nine patients in the POPS and ESWL groups, respectively. On comparing the POPS and ESWL groups, the treatment success rates were 78.9% (15 patients) in the POPS group and 70.2% (33 patients) in the ESWL group (*p* = 0.55). Among the four patients with treatment failure in the POPS group, we were unable to reach the pancreatic stone in two, one patient unexpectedly had a stone located in close proximity to the papilla, which resulted in the inability to secure treatment space, and one patient experienced pancreatic duct perforation during LL, leading to treatment discontinuation. Among these four patients, one was successfully converted to ESWL, and three were initially asymptomatic and required no additional treatment. In the ESWL group, all 14 patients with treatment failure exhibited a lack of fragmentation with ESWL alone. Additional treatments for the 14 patients with unsuccessful ESWL included surgery in 1, pancreatic duct stenting in 4 (with 3 patients subsequently undergoing surgery), symptom improvement despite the remaining pancreatic stones in 2, and no additional treatment in 8 patients (3 patients were asymptomatic and 5 desired conservative management, including pain relief and observation). 

### 3.3. Procedure-Related Complications

Complications related to the procedures were observed in 21% (4 patients) of the POPS group (pancreatic duct perforation in 1, post-ERCP hyperamylasemia in 3 patients) and 6.3% (3 patients) of the ESWL group (mild post-ERCP pancreatitis in 1, bleeding due to EPST in 1, and bleeding due to balloon dilation of MPD stricture in 1 patient) (*p* = 0.09) (Table 3). Pancreatic duct perforation caused by the laser lithotripsy was successfully treated through surgical drainage of the pancreatic fistula. Eventually, all the patients who experienced complications recovered. 

### 3.4. Number of Sessions

The median total number of sessions was 1 (1–4) for the POPS group and 5 (2–28) for the ESWL group (*p* < 0.01) (Table 3). 

### 3.5. Multivariate Analysis for Treatment Success

Multivariate analysis to assess the factors influencing treatment success did not yield any significant results. The choice between POPS-L and ESWL did not affect treatment success (odds ratio: 1.43, 95% CI: 0.402–5.958) (Table 4). 

### 3.6. Long-Term Outcomes

After successful treatment, the stone recurrence rate was 26% (4/15), with a median observation period of 176 days (range: 0–1507 days) in the POPS group and 24% (8/33), with a median observation period of 329 days (range: 0–4171 days) in the ESWL group (*p* = 1.00) (Table 5). The cumulative recurrence rates did not show a significant difference (log-rank test, *p* = 0.53) (Figure 2). Additionally, a Gray test was conducted to reduce the impact of death events that could act as competing risks for recurrent pancreatic stones. However, no significant difference was observed between the two groups (*p* = 0.668). Among the four patients with recurrence in the POPS group, two remained asymptomatic and required no additional treatment, one underwent endoscopic stone removal, and one required additional EHL. Among the eight patients with recurrence in the ESWL group, all the patients underwent repeat ESWL and endoscopic treatment. 

## 4. Discussion

In the comparative analysis between POPS and ESWL for pancreatic duct stones in this study, the treatment success rates were 78.9% (15/19) vs. 70.2% (33/47) (*p* = 0.55), and the incidence of procedure-related complications was 21% (4/19) vs. 6.3% (3/47) (*p* = 0.09), indicating no significant differences in treatment success or safety. However, the POPS group required significantly fewer sessions than the ESWL group.

The evaluation of POPS and ESWL treatments for pancreatic stones has been primarily conducted in single-arm studies. POPS treatment for pancreatic stones was first reported in 1991 using laser lithotripsy [10], followed by the introduction of the treatment of patients using EHL in 1992 [11]. Subsequent reports, before the advent of the current disposable POPS tools, indicated stone clearance rates of 43–100% and complication rates of 0–28% [9]. However, these treatment outcomes were based on a small number of patients and indicated considerable variability, with challenges related to the usability and durability of POPS tools. This prevented the widespread adoption of POPS in clinical practice. In recent years, the introduction of disposable pancreatic and biliary endoscopes, such as the SpyGlass DS, has allowed for single-operator procedures, improved maneuverability with four-way angulation, and enabled irrigation and suction while keeping the lithotripsy device loaded, contributing to precise fragmentation with clear visualization. These advancements have been reported to be beneficial in the treatment of stones, particularly in challenging cases of biliary stones [12]. The availability of disposable POPS tools has significantly improved maneuverability, enabling the performance of procedures even in narrow and tortuous lumens, such as the pancreatic duct. Studies using disposable POPS tools (SpyGlass DS) have reported treatment success rates of 88.2–89.9% and adverse event rates of 4.7–10.1%, demonstrating improved treatment efficacy compared with traditional pancreatoscopy [13,14]. In our investigation, the use of disposable POPS tools (SpyGlass DS) resulted in a stone clearance rate of 78%, yielding favorable outcomes comparable to those reported previously. 

Although reports on the utility of POPS are increasing, comparative studies of POPS and ESWL are lacking. In the current study, we compared POPS-L with ESWL for pancreatic stones and observed that stone clearance, complications, and recurrence rates were comparable between the two methods. Moreover, the POPS treatment was superior in terms of reduced session frequency. A retrospective comparative study by Bick et al. [15] that assessed the treatment outcomes of POPS-L and ESWL for pancreatic stones reported similar results. In their study, POPS-L was performed in 18 patients, while ESWL was performed in 240 patients, with a clearance rate of 88.9% vs. 86.7% (*p* = 1.00), the number of total procedures at 1.6 ± 0.6 vs. 3.1 ± 1.5 (*p* < 0.001), a procedure time of 101.6 ± 68.2 vs. 191.8 ± 111.6 min (*p* = 0.001), and complication rates of 5.6% vs. 6.3% (*p* = 1.000), which were similar to our findings. The reasons for the lower number of treatment sessions with POPS may be attributed to the following: (1) POPS allows for both fragmentation and removal within a single session, in contrast to ESWL, where separate endoscopic sessions are often required; and (2) POPS enables accurate targeting under direct visualization, facilitating efficient energy transmission to the stone, thus achieving effective fragmentation in a shorter time. 

Considering the main factors that contribute to treatment failure with POPS and ESWL, it is evident that, in the patients treated with POPS, the most common reason for treatment failure is the inability of the POPS tool to reach the pancreatic stone due to bending or narrowing of the pancreatic duct [13,14,16,17]. In our study, difficulty in the POPS tool insertion was the most common cause of technical failure (10.5% (2/19)). Thus, the success rate of POPS treatment is significantly influenced by the feasibility of the POPS tool insertion. Therefore, the shape of the pancreatic duct, which greatly influences the insertability of the POPS tool, may be a significant factor in treatment selection. In contrast, for ESWL, the primary factor for treatment failure was difficulty in the fragmentation of the pancreatic stones. ESWL is known to be unsuccessful or requires an increased number of treatment sessions in patients involving stones with high CT values, stones located in the pancreatic tail, large or multiple stones, and pancreatic duct narrowing [8,18,19]. However, the current study indicated no factors related to unsuccessful treatment with ESWL. 

Regarding long-term outcomes, the recurrence rates of pancreatic stones with POPS and ESWL were comparable at 26.6% (median observation period 176 days, range 0–1507 days) and 24.2% (median observation period 329 days, range 0–4171 days), respectively, and the cumulative recurrence rates were similar. However, the study by Bick et al. did not include a comparison of the recurrence rates for both treatments [15]. Previous reports have indicated that the recurrence rate after POPS-L ranges from 0% to 10.1%, while the recurrence rate after ESWL is reported to be 18.84% (95% CI, 15.83–20.40) [8,14,17]. These reports suggest a slightly higher recurrence rate with ESWL; however, our study found similar recurrence rates for both treatments. Unlike POPS, ESWL does not allow for the direct visualization of residual stones within the pancreatic duct, which may lead to missed detection of the remaining stones and contribute to recurrence. In our study, among four patients with recurrence after POPS treatment, two patients had stones that initially remained in the branch ducts and subsequently migrated to the main pancreatic duct. This suggests a distinct mechanism of recurrence specific to the POPS-L technique, in which the treatment of stones in the branch ducts is challenging. However, our study was a retrospective analysis involving a small number of patients, and further investigations are required to better understand these results. 

In terms of safety, the incidence of complications for POPS-L and ESWL was 21% and 6.3%, respectively. The breakdown of complications was as follows: for POPS-L, post-ERCP hyperamylasemia in three patients (15%) and main pancreatic duct (MPD) perforation in one patient (5%); and for ESWL, post-ERCP pancreatitis in one patient (2.1%) and bleeding after ERCP in two patients (4.2%). A recent systematic review and meta-analysis reported a complication rate of 14.09% (95% CI 8.31–22.90) for POPS, with pancreatitis being the most frequently reported complication, at 8.73% (95% CI 4.50–16.27) [20]. In our study, post-ERCP hyperamylasemia was observed in three patients, which was the most common complication. With POPS-L, mechanical irritation from the POPS tool insertion, along with the possibility of increased intraductal pressure due to the saline infusion to maintain visualization, may contribute to pancreatitis, even in patients with chronic pancreatitis. For ESWL, the recent systematic review and meta-analysis reported a post-ESWL pancreatitis rate of 4.0% (95% CI 2.5–5.8) and a post-ESWL cholangitis rate of 0.5% (95% CI 0.2–0.9), with no reported deaths [8]. Although the frequency is low, attention should be paid to potential injuries to the surrounding organs, such as subcutaneous hematoma, submucosal gastric hematoma, hematuria following renal injury, liver injury, lung injury, and vascular injury [8,21,22,23]. However, in our study, no complications were observed with ESWL, although a few complications related to the endoscopic procedure occurred in the ESWL group. Both a previous report [15] and our study observed comparable rates of complications for both treatments, although a slightly higher trend was observed for POPS-L. Pancreatitis is a common complication for both treatments [8,20], emphasizing the need for caution, particularly with POPS-L, in which elevated intraductal pressure and the potential for pancreatic injury tend to occur. 

The treatment outcomes of POPS and ESWL for pancreatic stones might be comparable, with POPS showing superiority in terms of the number of treatment sessions. The choice between POPS and ESWL treatment is considered to depend on the characteristics of the individual patient. In patients in whom successful access of the POPS tool to the targeted pancreatic stone is anticipated, such as those featuring a main pancreatic duct with minimal bends or constriction from the papilla to the stone, and in instances where the stone is confined within the main pancreatic duct, POPS may be the preferred initial option. This approach results in a more efficient treatment process, with fewer sessions than ESWL. However, cases where the insertion of a POPS tool is predicted to be challenging, such as in patients with a tortuous or narrow main pancreatic duct, ESWL may be prioritized. Hence, more efficient treatment strategies may be developed considering these characteristics and selectively using POPS and ESW. Therefore, further studies in this direction are warranted. 

This study had several limitations. Firstly, it was a retrospective analysis conducted on a small cohort of patients treated across three tertiary care centers, restricting the areas that could be explored due to its retrospective data extraction. Moreover, it potentially introduced significant selection bias due to institutional discretion. However, the preference for POPS treatment was primarily driven by patient preferences for rapid intervention, irrespective of the stone properties. Secondly, due to variations in the number of ESWL shots per session across different facilities or countries, it is slightly challenging to globally assess the treatment outcomes of ESWL in terms of the number of sessions required. Thirdly, some cases included in this study had no symptoms arising from pancreatic stones, although treatment for asymptomatic pancreatic stones is basically not recommended in guidelines [1,2]. Thus, this study focused on stone clearance capability, not on symptom relief. Finally, POPS treatment was prioritized based on patient preferences in this study, despite several guidelines recommending ESWL as the first choice for large pancreatic duct stones [1,2]. However, the innovative progress of POPS may alter the existing algorithm for pancreatic stone management. Hence, further evaluation through randomized controlled trials is required. 

## 5. Conclusions

Both ESWL and POPS-L are effective and safe methods for treating pancreatic duct stones. However, POPS-L may potentially reduce the number of treatment sessions required.

## Figures and Tables

**Figure 1 diagnostics-14-00891-f001:**
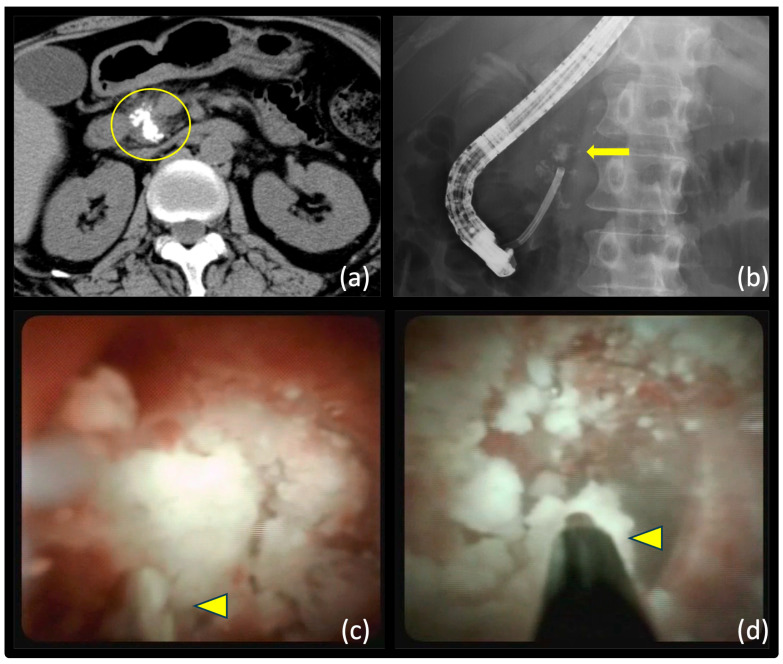
Peroral pancreatoscopy-guided lithotripsy. (**a**) The patient has a main pancreatic duct stone in the pancreas head (yellow circle). (**b**) POPS tool is inserted towards the main pancreatic duct stone (yellow arrow) through the mother scope. (**c**) EHL tool (yellow arrowhead) is inserted into the front of the pancreatic stone, followed by stone lithotripsy. (**d**) Pancreatic stone is finely fragmented by EHL (yellow arrowhead) and spontaneously expelled through the papilla. POPS, peroral pancreatoscopy; POPS-L, peroral pancreatoscopy-guided lithotripsy; EHL, electrohydraulic lithotripsy.

**Figure 2 diagnostics-14-00891-f002:**
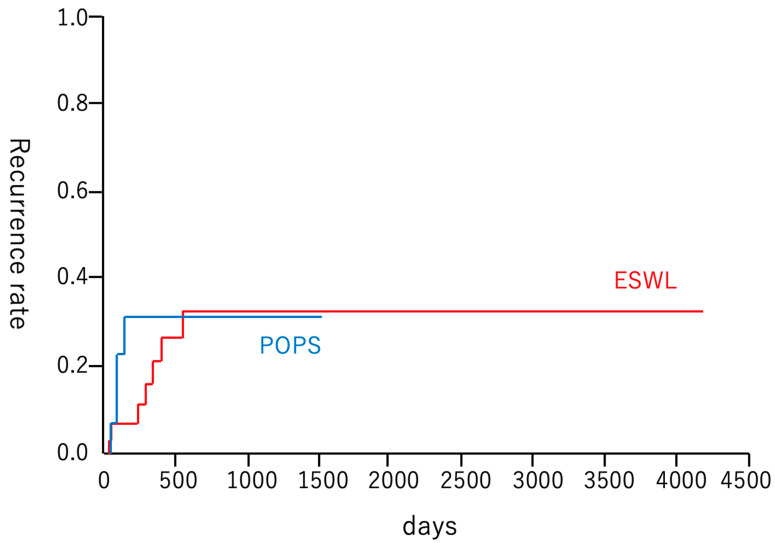
Kaplan–Meier curve for main pancreatic duct stone recurrence. The red and blue lines indicate the cumulative rates of main pancreatic duct stone recurrence in the ESWL and POPS groups, respectively. The log-rank test demonstrated no significant difference between the two groups (*p* = 0.532). ESWL, extracorporeal shock wave lithotripsy; POPS, peroral pancreatoscopy.

**Table 1 diagnostics-14-00891-t001:** Patient characteristics.

	Total *n* = 66	POPS-L *n* = 19	ESWL *n* = 47	*p*-Value
Median age, years (range)	64 (22–86)	62 (41–86)	65 (22–80)	0.744
Male, *n* (%)	52 (78)	14 (73.6)	38 (80.8)	0.522
Etiology of chronic pancreatitis, *n* (%)				0.531
Alcohol-related	58 (87.8)	15 (93.7)	40 (85.1)	
Idiopathic	7 (10.6)	1 (6.0)	6 (12.7)	
Autoimmune	1 (1.5)	0	1 (2.1)	
Symptom, *n* (%)				0.406
Pain	21 (31.8)	4 (21.0)	17 (36.1)	
Pancreatitis attack	32 (48.4)	9 (47.3)	23 (48.9)	
Pseudocyst-related	2 (3.0)	1 (5.2)	1 (2.1)	
No symptoms (anticipation of preserving pancreatic function)	11 (16.6)	5 (26.3)	6 (12.7)	

POPS-L, peroral pancreatoscopy-guided lithotripsy; ESWL, extracorporeal shock wave lithotripsy.

**Table 2 diagnostics-14-00891-t002:** Characteristics of pancreatic stones and pancreas.

	Total *n* = 66	POPS-L *n* = 19	ESWL *n* = 47	*p*-Value
Median diameter of stone, mm (range)	10 (5–40)	10 (5–21)	10 (5–40)	0.92
Median number of stones, *n* (%)	1 (1–3)	1 (1–3)	1 (1–3)	0.186
Stone location, *n* (%)				1.00
Head	51 (77.2)	15 (78.9)	36 (76.6)	
Body/tail	15 (22.7)	4 (21.0)	11 (23.4)	
Median CT value of stone density, HU (range)	1043 (43–3814)	832 (534–1375)	1069 (43–3814)	0.126
MPD stenosis, *n* (%)	28 (42.4)	11 (57.8)	17 (36.1)	0.168
Median diameter of downstream MPD of stone, mm (range)	3.1 (1.4–8)	3.15 (1.4–8)	3 (2.1–8)	0.639
Median diameter of upstream MPD of stone, mm (range)	7.5 (3–15)	8.5 (5–12.6)	7 (3–15)	0.055

POPS-L, peroral pancreatoscopy-guided lithotripsy; ESWL, extracorporeal shock wave lithotripsy; CT, computed tomography; HU, Hounsfield Units; MPD, main pancreatic duct.

**Table 3 diagnostics-14-00891-t003:** Treatment outcomes.

	POPS-L *n* = 19	ESWL *n* = 47	*p*-Value
Treatment success, *n* (%)	15 (78.9)	33 (70.2)	0.554
Cause of treatment failure, *n*			
Difficulty of POPS insertion near stone	2		
MPD perforation	1		
Difficulty due to site (near the Ampula)	1		
EHL/LL, *n*	13/4		
Median number of all sessions, *n* (range)	1 (1–4)	5 (2–28)	*p* < 0.01
Median procedure time, minutes (range)	72 (30–187)		
Median number of ESWL sessions, *n* (range)		3 (1–22)	
Median number of ESWL shots, *n* (range)		9000 (2000–70,000)	
ESWL alone/combination of ESWL and endoscopic extraction, *n*		17/30	
Complications, *n* (%)	4 (21.0)	3 (6.3)	0.098
Hyperamylasemia	3	0	
Pancreatitis	0	1	
MPD perforation	1	0	
Bleeding	0	2	

POPS-L, peroral pancreatoscopy-guided lithotripsy; ESWL, extracorporeal shock wave lithotripsy; MPD, main pancreatic duct; EHL, electrohydraulic lithotripsy; LL; laser lithotripsy.

**Table 4 diagnostics-14-00891-t004:** Predictors of pancreatic stone clearance.

	Univariate Analysis			Multivariate Analysis		
	Odds Ratio	95%CI	*p*-Value	Odds Ratio	95% CI	*p*-Value
Male	2.666	0.6291–18.419	0.196	1.814	0.366–13.535	0.483
Age ≥ 65	1.607	0.541–4.904	0.392	1.154	0.352–3.807	0.811
Stone diameter ≥ 10 mm	1.329	0.445–4.163	0.611	1.21	0.363–4.129	0.755
CT value of stone ≥ 1043 HU	1.87	0.624–5.906	0.264	1.524	0.450–5.331	0.497
Body/tail stone	1.461	0.395–4.969	0.554			
Stricture of downstream MPD	1.12	0.367–3.345	0.839			
Upstream MPD diameter ≥ 7.5 mm	0.676	0.222–2.008	0.481			
Alcohol abuse	1.142	0.234–8.349	0.876			
Symptomatic stone	0.612	0.158–2.626	0.49			
ESWL	1.59	0.475–6.337	0.463	1.439	0.402–5.958	0.582
Complications	1.075	0.143–5.562	0.935			

CT, computed tomography; HU, Hounsfield Units; MPD, main pancreatic duct; ESWL, extracorporeal shock wave lithotripsy.

**Table 5 diagnostics-14-00891-t005:** Long-term outcomes.

	Total *n* = 48	POPS-L *n* = 15	ESWL *n* = 33	*p*-Value
Median observation periods, days (range)	310 (0–4171)	176 (0–1507)	329(0–4171)	0.235
Recurrence of pancreatic stone, *n* (%)	12 (25.0)	4 (26.6)	8 (24.2)	1

POPS-L, peroral pancreatoscopy-guided lithotripsy; ESWL, extracorporeal shock wave lithotripsy.

## Data Availability

Data is contained within the article; further inquiries can be directed to the corresponding author.

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
