# Peer review of "Peroral Pancreatoscopy-Guided Lithotripsy Compared with Extracorporeal Shock Wave Lithotripsy in the Management of Pancreatic Duct Stones in Chronic Pancreatitis: A Multicenter Retrospective Cohort Study"

_diagnostics, 2024, doi:10.3390/diagnostics14090891_

Round 1
Reviewer 1 Report
Comments and Suggestions for Authors
In general, I think the paper is well-written, and most of the applied methods seem appropriate. I have only a few questions concerning some methodological aspects:
- In Table 3, it was not clear to me why the authors present a "median number of sessions" (line no. 7) and afterwards (line no. 9) a "median number of ESWL sessions". Where is the difference? Could you please clarify?
- In their (Kaplan-Meier) analysis of pancreatic duct stone recurrence, did the authors consider to apply a competing risk analysis? This could be important, because "death" consitutes a relevant competing risk in all models not involving mortality.
Author Response
Dear all editors and reviewers,
We are grateful for your valuable feedback and for considering our article for publication. We have diligently revised our manuscript, marking alterations in yellow and addressing each comment in our responses.
Firstly, I have removed comma at line 338 since it was unnecessary. (from ‘procedure, occurred’ to ‘procedure occurred’)
I appreciate your helpful feedback. Thanks to your first suggestions, I have made corrections from 'Median number of sessions' to 'Median number of all sessions' at line 7 in Table 3. 'Median number of all sessions' refers to the median number of all sessions, including both endoscopic sessions and ESWL sessions.
I also appreciate your valuable second suggestion. Following your comment, the following content has been added in the Results section (at line 231-233); Additionally, a Gray test was conducted to reduce the impact of death events that could act as competing risks for recurrent pancreatic stones. However, no significant difference was observed between the two groups (p=0.668).
Thank you so much again.
Reviewer 2 Report
Comments and Suggestions for Authors
The authors (Iwata et al,) examined the “Peroral pancreatoscopy-guided lithotripsy compared with extracoporeaal shock wave lithotripsy in the management of pancreatic duct stones in chronic pancreatitis, A multicenter retrospective cohort study”. It is very interesting and relevant to the current understanding of pancreatic duct stone development and endoscopic therapy.
But it will be very informative for the reader, if authors give more some deails of this study
(1) How many patients are diabetic, do the authors collected the plasma glucose level?
(2) As the authors described, the sample size of these studies were very limited, so the authors take caution in explain the results.
Author Response
Dear all editors and reviewers,
We are grateful for your valuable feedback and for considering our article for publication. We have diligently revised our manuscript, marking alterations in yellow and addressing each comment in our responses.
Firstly, I have removed comma at line 338 since it was unnecessary. (from ‘procedure, occurred’ to ‘procedure occurred’)
point by point response to reviewer 2:
Regarding your first comment, I agree that it's a crucial point as diabetes are closely linked to pancreatic function and its long-term outcomes. However, the data on plasma glucose levels are unavailable for most of the study populations. We have changed the sentence in limitation section from 'Firstly, it was a retrospective analysis conducted on a small cohort of patients treated across three tertiary care centers, potentially introducing significant selection bias due to institutional discretion' to 'Firstly, it was a retrospective analysis conducted on a small cohort of patients treated across three tertiary care centers, restricting the areas that can be explored due to its retrospective data extraction. Moreover, it potentially introduced significant selection bias due to institutional discretion.' (at line 355-358) Since diabetes is directly related to pancreatic disease, we will consider your valuable suggestion for future research.
Regarding your later comment, I acknowledge that the number of patients treated with POPS for pancreatic stones has been limited in previous and current studies. Nonetheless, I believe that POPS treatment can be highly beneficial for certain patients. Further investigation into this treatment modality is warranted.